# Cytokines as Biomarkers for Evaluating Physical Exercise in Trained and Non-Trained Individuals: A Narrative Review

**DOI:** 10.3390/ijms241311156

**Published:** 2023-07-06

**Authors:** Paulina Małkowska, Marek Sawczuk

**Affiliations:** 1Institute of Physical Culture Sciences, University of Szczecin, 71-065 Szczecin, Poland; 2Doctoral School, University of Szczecin, 70-384 Szczecin, Poland

**Keywords:** cytokines, chemokines, physical activity, immune response, gene expression, inflammation, biomarkers

## Abstract

Physical activity and exercise training have numerous health benefits, including the prevention and management of chronic diseases, improvement of cardiovascular health, and enhancement of mental well-being. However, the effectiveness of training programs can vary widely among individuals due to various factors, such as genetics, lifestyle, and environment. Thus, identifying reliable biomarkers to evaluate physical training effectiveness and personalize training programs is crucial. Cytokines are signaling molecules produced by immune cells that play a vital role in inflammation and tissue repair. In recent years, there has been increasing interest in the potential use of cytokines as biomarkers for evaluating training effectiveness. This review article aims to provide an overview of cytokines, their potential as biomarkers, methods for measuring cytokine levels, and factors that can affect cytokine levels. The article also discusses the potential benefits of using cytokines as biomarkers, such as monitoring muscle damage and inflammation, and the potential for personalized training programs based on cytokine responses. We believe that the use of cytokines as biomarkers holds great promise for optimizing training programs and improving overall health outcomes.

## 1. Introduction

Physical exercise refers to any physical activity that is planned, structured, and performed with the intention of improving or maintaining physical fitness and overall health [1]. The benefits of physical exercise are numerous and wide-ranging, with both physical and mental health improvements being reported [2]. Regular physical exercise has been shown to improve cardiovascular health, leading to a reduced risk of heart disease, stroke, and high blood pressure [3]. Physical exercise can also improve bone density, reducing the risk of osteoporosis and fractures [4]. Furthermore, regular physical activity can help to control weight [5], improve insulin sensitivity [6], and reduce the risk of type 2 diabetes [7]. Physical exercise also has numerous benefits for mental health. It has been shown that physical exercise reduces symptoms of depression and anxiety [8], improves mood and cognitive function [9], and reduces the risk of cognitive decline and dementia [10]. It also can improve sleep quality and reduce stress levels [11].

There are many different types of physical exercise, including aerobic exercise, strength training, and flexibility exercises [12]. Aerobic exercise, such as jogging, cycling, or swimming, is great for improving cardiovascular health and burning calories [13]. Strength training, such as weight lifting or resistance band workouts, can improve muscle strength and help to maintain bone density [14]. Flexibility exercises, such as yoga or stretching, can help to improve mobility and reduce the risk of injury [15]. However, the intensity and duration of exercise can vary greatly among individuals, and the effectiveness of exercise programs can be difficult to assess. 

With the growing popularity of physical exercise, there is an increasing need for reliable biomarkers that can be used to evaluate the effectiveness of training programs. Biomarkers are measurable indicators of biological processes, and they can provide valuable insights into the effects of physical exercise on the body [16]. Reliable biomarkers are important because they can help to identify the optimal training program for a specific individual or population. By measuring biomarkers before and after exercise, it is possible to determine which exercises are most effective at achieving specific health outcomes. Additionally, biomarkers can be used to track progress over time, allowing individuals to see the benefits of their hard work.

Currently, there are a variety of different biomarkers that are being used to evaluate physical exercise effectiveness. These include physiological biomarkers, such as heart rate, blood pressure, and VO_2_max, as well as biochemical biomarkers, such as cytokines, hormones, and metabolic markers [17,18,19]. While these biomarkers are useful, there is a need for more reliable and accurate biomarkers that can provide more specific information about the effects of exercise on the body. One of the most promising biomarkers for evaluating training effectiveness is cytokines [19]. Cytokines are signaling molecules that are involved in immune and inflammatory responses, and they play an important role in regulating the body’s response to exercise [20]. By measuring cytokine levels before and after exercise, it is possible to determine the effectiveness of different types of exercise at eliciting specific immune and inflammatory responses.

This review article aims to provide an overview of the current research on the use of cytokines as biomarkers for evaluating training effectiveness. We will discuss the different types of cytokines that have been studied in relation to exercise, their physiological roles, and how changes in their levels can be used to assess the effectiveness of training programs. We will also examine the potential limitations and challenges associated with using cytokines as biomarkers, as well as future research directions in this field. Overall, this review will highlight the potential of cytokines as a valuable tool for monitoring and evaluating the physiological response to exercise and guiding the design of effective training programs.

## 2. Cytokines in Physical Activity

Cytokines are a diverse group of signaling molecules that play important roles in immune and inflammatory responses, as well as in the regulation of cellular growth and differentiation. There are many different types of cytokines, each with unique functions and effects on the body, such as interleukins (ILs), tumor necrosis factor (TNF), interferons (IFNs), chemokines, colony-stimulating factors (CSFs), and growth factors. Overall, cytokines are a diverse and important group of signaling molecules that play crucial roles in the immune system, inflammation, and cellular growth and differentiation. The different types of cytokines have unique functions and effects on the body, and their dysregulation can contribute to the development of a variety of diseases [21]. In recent years, research has increasingly focused on the role of cytokines in physical activity and exercise, as exercise has been shown to modulate the expression and secretion of cytokines in various tissues and organs.

### 2.1. Interleukins (ILs)

Interleukins (ILs) are a group of proteins involved in communication with leukocytes by binding to specific receptors. ILs are mainly synthesized by monocytes, macrophages, CD4+ helper T cells, and endothelial cells [21].

The IL-1 family of pro-inflammatory cytokines consists of 11 members. They take part in promoting the activity of cells of the innate immune system and activating and enhancing the function of T lymphocytes [22]. IL-1 up-regulates a wide variety of genes, including those responsible for coding cytokines, thereby increasing its own expression, as well as that of IL-2 and IL-6 [23]. IL-1 has pro-inflammatory and degenerative effects on the joint surface [24] and contributes to cartilage degeneration. Together with TNFα, it promotes fat catabolism [25].

In the context of the effects of exercise on cytokine secretion, several reports can be found on IL-1. Studies have shown that trained runners after a distance of 20 km do not show increased levels of IL-1β immediately after running. However, significantly increased levels of IL-1β are found 1 h after running and persist until the next day [26]. The situation is different for IL-1ra, whose increase is observed immediately after exercise. IL-1ra is a receptor antagonist for IL-1 that can participate in feedback regulation by binding to receptors for IL-1, blocking them, and preventing the action of both IL-1α and IL-1β. Trained long-distance runners showed lower resting concentrations of IL-1ra by 7.8–33% than after the treadmill test. However, 1 h after the test, their levels dropped significantly [27]. A higher increase was observed in ultramarathon runners after running 90 km, where IL-1ra increased 6.1-fold [28]. Slightly different results are observed during bicycle ergometer tests. An increase in IL-1ra is observed during 1 h of recovery, where it reaches a peak concentration and then decreases and returns to normal after 24 h [29]. IL-18, which also belongs to the IL-1 family, also appears to be associated with physical activity. Trained cyclists had unchanged plasma levels of IL-18 after a distance of 230 km; however, after 24 h, the levels dropped significantly (about 50%). In contrast, calculated concentrations of free IL-18 revealed a 32–68% decrease after exercise. The authors suggest that such a decrease in free IL-18 may contribute to increased susceptibility to infection in athletes undergoing exhaustive exercise [30]. Other studies also show a correlation between IL-18 levels and exercise. A 43% decrease in IL-18 levels after aerobic exercise training in inactive men and women with metabolic syndrome [31] and a 20% decrease in IL-18 mRNA in adipose tissue after exercise training in obese individuals have been demonstrated [32].

Another described cytokine is IL-2, which is a growth factor for T lymphocytes. It is mainly produced by CD4+ and CD8+ T lymphocytes but can also be expressed by dendritic and NK cells [21]. This cytokine has a feedback effect on the immune response. When a T lymphocyte is stimulated, it induces the appearance on its surface of molecules that enable apoptosis of this cell. IL-2 also promotes the suppression of inflammatory responses [33]. 

Studies have shown that physical activity is associated with changes in IL-2 secretion. The runners had significantly reduced IL-2 concentrations directly after the 5 km race, but after 24 h they were significantly increased compared to pre-exercise values [34]. Other studies on runners confirm a 32% or 55% decrease in IL-2 concentrations directly after the race [35,36]. However, the researchers noted a significant increase in IL-2 levels in women with sedentary lifestyles after a 6-week program of aerobic dance exercise [37]. Slightly different observations were made in a group of professional rowers. The initial study showed a significant (about 4-fold) increase in IL-2 after a 2 km rowing ergometer test and a return to normal after 24 h. In contrast, after a 6-week training program, the same rowers had no significant difference in IL-2 levels after the same test, and a 3-fold decrease was noted after 24 h [38]. These results may suggest that changes in IL-2 secretion depend on the regularity and intensity of exercise.

IL-4 is a cytokine responsible for regulating allergic conditions and activating the immune response against extracellular parasites. It is produced mainly by Th2 cells, basophils, eosinophils, and mast cells [21].

Only a few studies have included IL-4 among the parameters studied. Most studies found no significant differences in IL-4 levels in runners [35,36,39,40,41]. The lack of significant changes in IL-4 concentrations is also confirmed by studies on female rowers performing a test on a rowing ergometer [42]. However, Skinner et al. performed a study in which they compared the results obtained after a 40 km and 171 km race. In the case of the 40 km race, there were also no significant differences in IL-4 levels, but IL-4 concentrations increased significantly after the 171 km race [43]. 

The best-studied and described pro-inflammatory cytokine in physical activity is IL-6. It is produced by not only immune cells (T and B lymphocytes, NK cells, and monocytes) but also non-immune cells (glial cells, smooth muscle cells, chondrocytes, and astrocytes) [23]. IL-6 is a potent mediator of the acute phase, which is a cascade of responses initiated by stressors, such as inflammation, tissue damage, or heavy physical activity. This mechanism is designed to prevent further damage and initiate repair processes [44]. Despite the well-characterized pro-inflammatory role of IL-6, its role has been shown to be different in the context of physical activity. IL-6 participates in anti-inflammatory effects through the induction of anti-inflammatory cytokines [45]. Molecular studies have shown that this dual action of IL-6 is dependent on cell signaling. A soluble IL-6 receptor is involved in the pro-inflammatory effect, while Gp130 receptors are involved in the anti-inflammatory effect [46].

The vast majority of studies to date show an increase in IL-6 concentrations; however, the magnitude of this increase depends on exercise duration, intensity, and mental attitude [47,48]. A study performed on athletes after the Brussels Marathon showed as much as a 45-fold increase in plasma IL-6 concentrations immediately after exercise and 1 h afterward [49]. Another 42 km marathon resulted in a 100-fold increase in IL-6 concentrations in young healthy runners [35], and ultramarathon runners showed a 50.2-fold increase in IL-6 after a 160 km run [28]. Mean IL-6 concentrations after 1 h of submaximal exercise on a rowing ergometer in women and after a 30 min rest were also significantly higher. The coefficient of variation was 0.47 [42]. In elite male rowers after a submaximal 90 min session on the ergometer, maximum IL-6 concentrations occurred 2 and 3 h after the test. It was 7.5 times higher than the pre-exercise concentration [50]. Studies also suggest that an increased level of training may contribute to a lower IL-6 resting level. After 4 months of dance training (twice a week, 25 min of Zumba dance + 25 min of Brazilian folk dances), there was a 60% decrease in IL-6 in healthy women. The samples were taken before the start of training and 72 h after the last scheduled training. Moreover, it should be emphasized that this study, compared to those previously described, was conducted on non-trained individuals [51]. There are also data on the lack of significant changes in IL-6 levels in men after 6 weeks of training on a bicycle ergometer [52].

There are also few data on the increase in IL-6 in skeletal muscles. A study in Wistar rats using an animal model of overtraining showed a 2.17-fold increase in IL-6 gene expression in gastrocnemius calf muscle 36 h after the last session. Expression levels decreased after a 1-week recovery period [53]. An increase in intramuscular protein levels of IL-6 was also observed in mice in response to the excessive downhill running protocol [54,55].

Because IL-6 is produced locally in working skeletal muscle, it was initially thought that the increase in this cytokine might be related to muscle damage, as creatine kinase (CK) levels were also increased. However, further research in this area has contradicted this theory. The huge increase in plasma IL-6 levels in exercise models, in which CK levels are unchanged or only increased a few times, is related to mechanisms other than muscle damage [56].

There are data that highlight the importance of an increase in IL-6 concentrations in experiencing fatigue, pain, mood changes, or concentration disturbances characteristic after physical activity [57]. A study involving the administration of antibodies against IL-6 to athletes covering a distance of 10 km resulted in a reduction in the feeling of fatigue. Moreover, when recombinant IL-6 was administered to healthy people at rest, it caused them to feel the fatigue characteristic of physical activity [58].

Another interleukin that may play a role in the immune response during physical activity is IL-10. This is also known as human cytokine synthesis inhibitory factor (CSIF) and is an anti-inflammatory cytokine. IL-10 is a cytokine with pleiotropic effects in immunoregulation and inflammation. It is secreted mainly by antigen-presenting cells, such as activated T cells, monocytes, B cells, and macrophages [59]. It reduces the expression of MHC-II on the surface in monocytes without affecting MHC-II transcription or translation [60]. 

Studies have shown that physical activity can increase the production of IL-10, which has important implications for recovery and performance. A 90 km run causes a 9.5-fold increase in IL-10 concentration in ultramarathoners [28], while a 42 km run raised IL-10 concentrations 3.5 times [35]. However, the mean IL-10 concentrations after 1 h of submaximal exercise on a rowing ergometer in women and after a 30 min rest were non-significantly reduced [42]. The results of many studies clearly indicate that IL-10 production induced by physical activity is inconclusive. Both increases and decreases and no changes in the concentration of this interleukin are observed. Despite this ambiguity, it could be observed that IL-10 levels are highest during recovery from physical activity. However, this increase is influenced by active muscle mass and exercise intensity [61]. The meta-analysis showed that the most important determinant of the magnitude of the increase in IL-10 concentration caused by physical activity is its duration [62]. The increase in IL-10 is thought to prevent potentially harmful chronic inflammation and tissue damage associated with physical activity [63]. It is likely that the increase in IL-10 is associated with an increase in IL-6. This was demonstrated in a study where the administration of recombinant IL-6 increased IL-10 and IL-1ra in young healthy volunteers [64]. The increased secretion of IL-10 during intense exercise may be responsible for moving the immune response in an anti-inflammatory direction by inhibiting the secretion of pro-inflammatory cytokines, such as IL-1, TNF-α, IL-8, and MIP [65]. 

Other cytokines belonging to the interleukin group are poorly studied, and there are only isolated reports regarding their potential role in the immune response during physical activity. IL-12, which is produced by macrophages, consists of two subunits: p35 and p40. The p35 subunits form the p70 subunit, while the p40 subunits can form a homodimer with antagonistic properties to IL-12. IL-12 p70 is a promoter of cellular immunity, and IL-12 p40 has the opposite properties. Studies have shown a significant increase in IL-12 p40 after a test performed on a treadmill [66]. Another interleukin that seems to play an important role during physical activity is IL-15. It acts as an immunoregulatory mediator and is strongly expressed in skeletal muscle after exercise [67]. It has been shown to contribute to increased myosin production in skeletal muscle [68]. This mechanism occurs because of its anabolic functions, which also play a role in fat reduction [69].

A summary of information on the effects of physical activity on interleukins secretion is presented in Table 1.

### 2.2. Chemokines

Chemokines are a large family of structurally homologous cytokines. Their function is to stimulate the movement of leukocytes and regulate their migration from the blood to tissues in the process of chemotaxis. In addition, they control homeostatic immune cells, such as neutrophils, B lymphocytes, and monocytes, by moving between bone marrow, blood, and peripheral tissues. This is the basis for classifying them as chemotactic cytokines [21,70]. Chemokines are a numerous group, but two main families can be distinguished within them: the CCs and CXCs. Chemokines that belong to CC are chemotactic against monocytes and a small subset of lymphocytes, while CXC chemokines are more specific against neutrophils [21]. 

The best-known chemokine involved in the immune response to exercise is IL-8 (CXCL8). It belongs to the CXC chemokine family and is responsible for recruiting neutrophils and sustaining the inflammatory response. It is produced by endothelial cells, fibroblasts, monocytes and macrophages, neutrophils, and T lymphocytes. It can be increased in expression by various stimuli, including IL-1 [71]. In the context of exercise, IL-8 is produced locally in the muscles during exercise, and an increase in its concentration is observed only after intense exercise [67]. After the marathon run, peak IL-8 values were recorded 0.5 h after finishing the run and were 6.7-fold higher than resting values [72]. In the case of the ultramarathon (160 km), the IL-8 concentration value immediately after the run was 2.5-fold higher than the resting value [28]. A significant increase in IL-8 mRNA was observed in cyclists after 2.5 h of cycling. An almost 30-fold increase in IL-8 mRNA was observed in skeletal muscles obtained by biopsy immediately after exercise [73]. 

Another chemokine tested for physical activity is monocyte chemotactic protein-1 (MCP-1, chemokine nomenclature: C–C motif chemokine ligand 2 (CCL2)), which belongs to the CC family of chemokines [21]. A significant increase in MCP-1 concentration was observed after a 1 h submaximal rowing ergometer exercise in female rowers [42] and after the marathon race in male runners [74]. Experienced soccer players subjected to the Wingate Anerobic Test (WAnT) showed minimal MCP-1 mRNA expression as measured immediately after the test. This was followed by a continuous increase over time with a maximum at the sixth hour after WAnT. In non-training subjects, the results were different. The number of MCP-1 transcripts was relatively high immediately after the test, and then the observed MCP-1 expression decreased to a lower level [75]. Also, when running on a treadmill for 45 and 60 min with VO2max = 60% or 80%, significant increases in this cytokine level were found in the plasma of trained male runners [66]. 

Macrophage inflammatory protein (MIP), a member of the CC family of chemokines, is also noteworthy. Its main representatives, i.e., CCL3 (MIP-1α) and CCL4 (MIP-1β), are produced by many cells, especially macrophages, cells, and lymphocytes. MIP-1 proteins are best known for their chemotactic and pro-inflammatory effects but can also promote homeostasis [76]. A study with male athletes showed that after a marathon run (42 km), MIP-1α and MIP-1β concentrations peaked 0.5 h after finishing and were 3.5-fold and 4.1-fold higher than resting levels, respectively [72].

A summary of information on the effects of physical activity on chemokines secretion is presented in Table 2.

### 2.3. Interferons (IFNs)

Interferons (IFNs) are a group of signaling proteins belonging to cytokines. They were first identified through their antiviral properties. IFNs not only have important antiviral effects but also have a role in antitumor and immunomodulatory responses [77]. They are used to communicate between cells in order to trigger the immune system’s defense mechanisms, which help fight pathogens [78]. There are three classes of IFNs based on the receptor through which they signal: type I (IFN-α, IFN-β, IFN-ε, IFN-κ and IFN-ω), which binds to a specific cell surface receptor complex IFN-α/β receptor (IFNAR); type II (IFN-γ), which binds to the interferon-gamma receptor (IFNGR); and type III (IFN-λ), which signals through a heterodimeric cell surface receptor composed of two chains: IFNLR1 and IL10RB [78,79,80,81].

Few papers address the effect of physical activity on IFNs. Even when studies have been conducted in this area, they have focused mainly on type II IFNs and the IFN-γ belonging to this group. It is known that IFN-γ is an anti-inflammatory cytokine. Studies on the effects of a single moderate-intensity exercise and one high-intensity exercise and one month of regular moderate exercise on IFN-γ have shown that its expression decreases after a single moderate exercise, further decreases after one high-intensity exercise, and increases at the end of a month of regular moderate exercise, which exceeds the baseline value. These results show that only regular physical activity has a beneficial anti-inflammatory effect [82]. These results were confirmed in a study of IFN-γ levels after two months of moderate-intensity training. A significant increase in IFN-γ was observed in samples taken after exercise compared to samples taken before exercise. IFN-γ levels decreased significantly after two months of rest [83].

### 2.4. Tumor Necrosis Factor (TNF)

Tumor necrosis factor alpha (TNF-α, also known as TNF superfamily member 2 (TNFSF2) or simply TNF) is a pluripotent cytokine that may be involved in metabolic disorders such as insulin resistance and is associated with type 2 diabetes [84]. TNF-α is mostly produced by macrophages and adipocytes. It is a major mediator of the acute inflammatory response. Conjugated with IL-6 and IL-1, it is involved in local inflammatory responses [85]. Its main physiological function is to stimulate leukocytes, signaling sites of inflammation, and activating them to eliminate microorganisms and reduce inflammation [86]. Two receptors are involved in TNF signaling: TNFR1 and TNFR2. TNFR1 is constitutively expressed on most cell types, and its signaling is mainly pro-inflammatory and apoptotic. TNFR2 is mainly restricted to endothelial cells, epithelial cells, and subsets of immune cells, and its signaling is anti-inflammatory and promotes cell proliferation [87,88].

As with most cytokines, the response of TNF-α to exercise largely depends on the intensity of training. Studies have shown that regular exercise causes TNF-α suppression through IL-6 stimulating the appearance of anti-inflammatory cytokines, such as IL-1ra and IL-10 [89]. A significant decrease in TNF-α was also observed in older women after resistance training, where both serum levels of the cytokine and expression of the gene encoding TNF-α significantly decreased immediately after exercise [86]. The results are different for acute exercise, such as marathon running. Running 42 km has been shown to cause a peak 2.3-fold increase in TNF-α concentrations in the first hour after exercise in adult men [48]. However, with 60 min of eccentric ergometer exercise, there were no significant differences in plasma TNF-α levels in either the young male or elderly groups. Nevertheless, the same study reported an increase in plasma sTNF-R1 levels in both groups after exercise. Peak levels of sTNF-R1 were reached at 1 h post-exercise in the young men’s group and were 1.4 times higher than resting levels and in the older men’s group at 2 h post-exercise and were 1.2 times higher than resting levels. Moreover, sTNF-R1 levels returned to pre-exercise values 3 h post-exercise in elderly subjects, while levels in young subjects were elevated until day 2 post-exercise [90].

It has been suggested that the increase in TNF-α levels with strenuous training is caused by muscles being damaged during exercise [91]. TNF-α expression in skeletal muscle increases during eccentric exercise, which leads to muscle damage, although no differences are found in resting mRNA levels for TNF-α [92].

### 2.5. Colony-Stimulating Factors (CSFs)

Colony-stimulating factors (CSFs) are glycoproteins which contribute to the growth of progenitors of monocytes, neutrophils, eosinophils, and basophils, as well as activating macrophages [93]. G-CSF (granulocyte-CSF) stimulates the formation of granulocyte colonies and has hematopoietic properties. Recently, it has been attributed a role as an important modulator of the immune response [94]. M-CSF (macrophage-CSF) is a factor that stimulates the formation of macrophage colonies. It increases the feeding properties of monocytes and macrophages and stimulates the production of cytokines (TNF, IFN, IL-1, and G-CSF) [95]. 

Both G-CSF and M-CSF levels were shown to be significantly increased in plasma and urine in male runners after a 42 km marathon [74]. A test of professional male cross-country skiers also indicated an increase in G-CSF after a treadmill exercise test to exhaustion. Peak plasma concentrations of this cytokine occurred immediately after exercise and just 1 h after it returned to its resting value [96]. The increase in G-CSF during vigorous exercise and the emerging evidence for treating skeletal muscle myopathy with G-CSF suggests that it is a potential mediator of skeletal muscle repair [97].

### 2.6. Growth Factors

A growth factor is a secreted biologically active molecule that can affect cell growth. They can act on specific receptors on the cell surface, which then transmit growth signals to other intracellular components. Granulocyte-macrophage colony-stimulating factor (GM-CSF) is an example of a cytokine growth factor, as it promotes the production of white blood cells. Examples of proteinaceous growth factors include vascular endothelial growth factor (VEGF), epidermal growth factor (EGF), and platelet-derived growth factor (PDGF) [98]. One of the better-characterized growth factors is transforming growth factor-β (TGF-β), which is secreted by macrophages and plays a major role in wound fibrosis [99]. Other growth factors worth mentioning in the context of physical exercise are growth differentiation factor 15 (GDF15) and fibroblast growth factor-21 (FGF-21). GDF15 can be expressed and secreted by many tissues and cell types in response to cellular stress. It is only in recent years that its significant role in the regulation of metabolism has been observed. mRNA of GDF15 is expressed in a wide variety of cells and tissues, such as the kidney, lung, pancreas, heart, skeletal muscle, adipose tissue, liver, gastrointestinal tract, and brain, with the highest expression in the placenta. A stressful situation that leads to an increase in circulating GDF15 is exercise [100]. FGF-21 is mainly secreted in adipose tissue and skeletal muscle. Muscle-specific FGF-21 acts as an important regulator of muscle growth, inflammation, whole-body metabolism, and premature aging [101]. As an anti-inflammatory agent, FGF-21 stimulates fatty acid oxidation, the production of ketone bodies, and inhibits lipogenesis [102].

Studies have shown that 60 min of eccentric exercise on an ergometer slightly increases TGF-β1 levels in a group of young men and in a group of elderly people [90]. Recreational cyclists were also shown to have higher resting TGF-β1 concentrations compared to those with sedentary lifestyles [103]. Also, such activity as non-brisk walking for 1 h increased plasma concentrations of both free and total TGF-β1 and total TGF-β2 [104]. Moreover, mRNA levels of growth factors are differentially expressed in adipose tissue under the influence of physical activity. This has been confirmed by studies in a rat model. TGF-β1 and PDGF-AA mRNA levels were reduced in untrained rats 3 h after an intense weight-bearing exercise session compared to resting results, but VEGF-A mRNA levels remained unchanged. Resting mRNA levels of TGF-β1, PDGF-AA, and VEGF-A were significantly higher in trained rats than in untrained rats. In trained rats, a significant decrease in mRNA expression of TGF-β1, PDGF-AA, and VEGF-A was observed 3 h after exercise [105].

In the case of GDF15 and FGF-21, there was a 4.2-fold and 20-fold increase, respectively, in the plasma of male runners in response to marathon running. This increase was recorded immediately after the run, and GDF15 and FGF-21 levels returned to normal after 48 h [106]. However, a 5-week endurance exercise program involving three cycle ergometer sessions per week was reported to reduce serum FGF-21 levels in elderly men [107]. The situation is similar for a study evaluating the effect of 12 weeks of simple resistance exercise on levels of, among other things, FGF-21 in patients with non-alcoholic fatty liver disease (NAFLD). During the study, the resistance exercise group performed two exercises (push-ups and squats) three times a week on non-consecutive days. After 12 weeks of training, there was an almost 2-fold decrease in serum FGF-21 levels in the patients [108]. From these data, we are able to conclude that the type of physical activity influences the differences in FGF-21 levels; however, more studies need to be conducted, and the study group standardized to draw more detailed conclusions. However, in the case of GDF15, it seems that its levels increase with many forms of exercise. This is confirmed by studies on healthy men in whom plasma GDF15 levels increased in response to an exercise test on a cycle ergometer. A first increase of 34% was observed already during exercise and another 2 h after the test was stopped, where the value was 64% above the resting value [109]. In addition, moderately trained men have been shown to increase plasma GDF15 levels by approximately 15% in response to both endurance (cycling) and resistance exercise. In trained triathletes, 4 h cycling resulted in a 5.3-fold increase in GDF15 compared to resting values. GDF15 values returned to normal after 24 h [110]. 

### 2.7. Summary of the Effects of Exercise on Cytokine Secretion 

The above literature review makes it impossible to draw unanimous and universal conclusions about the effects of physical activity on the immune response. Diverse study groups, the use of different tests, and measurement methods affect the differences in results. It is claimed that a single intense exercise is characterized by an increased release of pro-inflammatory cytokines, while an increased release of anti-inflammatory cytokines is a secondary phenomenon [111]. Most of the studies discussed on intense exercise, including marathon, ultramarathon, or long-distance cycling, seem to support this theory. Pro-inflammatory cytokines, such as IL-1β, IL-6, IL-8, MCP-1, MIP-1, and TNF-α, are increased in secretion during this type of exercise. The observed increase in anti-inflammatory cytokines, such as IL-1ra and IL-10, is probably a secondary phenomenon caused by an increase in TNF-α and IL-6. In addition, it is worth noting that IFN-γ, which has anti-inflammatory properties, undergoes reduced secretion after a single high-intensity exercise. 

The second theory, which is supported by some studies, concerns moderate regular exercise. It is believed that this type of physical activity is characterized by decreased secretion of pro-inflammatory cytokines and increased secretion of anti-inflammatory cytokines [111,112]. Most of the studies involve intense exercise; so as of today, there are only a few studies that can confirm this finding. Decreased levels of pro-inflammatory cytokines, such as IL-18 and IL-6, have only been reported in studies examining the effects of regular training programs of several weeks on the immune response. Also, the decrease in TNF-α observed in older women after resistance training seems to support this theory. Due to the decrease in the above pro-inflammatory cytokines, the increase in anti-inflammatory cytokines, such as IL-1ra and IL-2, is not considered a secondary phenomenon. Moreover, anti-inflammatory IFN-γ undergoes a significant increase after a month of regular moderate exercise. The above results are shown graphically in Figure 1.

## 3. Effect of Training Level on the Amount of Secreted Cytokines 

The effect of training level on the amount of secreted cytokines is an important area of research, as it can help to identify how different levels of training affect the immune and inflammatory responses to exercise. Only a few studies have examined the relationship between training level and cytokine secretion. For example, one study found that trained athletes had lower levels of chemokine CCL2 compared to non-trained individuals, indicating a potential adaptation to chronic exercise [75]. Another study found that trained athletes had lesser magnitude of change in the cytokine levels (IL-6 and TNF-α), during strenuous exercise compared to untrained individuals [113]. Trainees also show a 2-fold increase in IL-8 levels during dynamic exercise, which is not observed in non-trainers [114]. 

However, other studies have shown conflicting results, with some indicating that training status may not have a significant impact on cytokine secretion. For example, a study of female rowers found no significant differences in cytokine levels between trained and untrained individuals. Plasma cytokine concentrations (IL-6, TNF-α, and IL-1ra) did not differ significantly between those two groups [115].

Overall, more research is needed to fully understand the relationship between training level and cytokine secretion, as well as how this relationship may vary depending on the type of exercise and individual factors, such as age, sex, and overall health status. Nevertheless, it is clear that cytokine biomarkers can provide valuable information about the immune and inflammatory responses to exercise, which can help coaches and trainers to develop effective training programs and prevent injury.

## 4. Cytokines as Biomarkers in Sport and Exercise

Recent research has highlighted the potential of using cytokines as biomarkers in various new areas beyond the traditional fields of immunology and inflammation. Some of these emerging areas include sports medicine, neurology, and psychiatry [116]. Cytokines have been suggested as potential biomarkers for evaluating training effectiveness in athletes and fitness enthusiasts. However, the pleiotropic nature of some cytokines makes it impossible to obtain precise information about an athlete’s health status by measuring individual cytokines. Therefore, it is suggested that in order to use cytokines as biomarkers in sports, one should select which ones are suitable for this role and in which groups they should be analyzed [19]. However, this is not an easy task for a number of reasons. One of them is to competently determine what information we are able to obtain from selected groups of cytokines. Cytokines, as signaling molecules, communicate causing one cytokine to be responsible for many phenomena occurring within the immune system (Figure 2). Based on the research conducted to this day (described in Section 2), we are able to suggest several potential applications for individual cytokines. 

One example for the use of cytokines as biomarkers is the identification of overtraining syndrome (OTS) in athletes. This is based on the ppendhypothesis that injuries to the muscular, skeletal, or joint systems are the initiator of OTS. The injury in question refers to situations such as continued training after an acute injury without an adequate recovery period, which can lead to an exacerbation of the original condition, and the progression of a “naturally” occurring exercise-related injury (AMT, adaptive microtrauma). It is assumed that the occurring injury causes the release of inflammatory factors, such as cytokines. The hallmark of OTS is intense training and limited rest, which causes chronic inflammation. If this is the case, the pro-inflammatory cytokines IL-1β, IL-6, and TNF-α can be biomarkers to identify OTS [117] but can also be used as indicators of muscle damage and general inflammation [65].

Another proposal for determining health in athletes is to analyze IL-10 and TGF-β as biomarkers of recovery and adaptation to training [65,118]. However, for now, this is only a suggestion that has not been more widely studied and standardized. The basis for this assumption is the increase in levels of these cytokines mainly during the recovery period observed only in trained individuals. These assumptions can be extended to the analysis of pro-inflammatory cytokines, such as IL-1β, IL-6, and TNF-α. Theoretically, a decrease in their concentrations should indicate decreasing inflammation, which means that the body is adapting to training at a given level. This is a suggestion based on studies confirming that high-intensity interval training (aquarobics) had an effect on reducing IL-6 levels [119]. Nonetheless, it would need to be further investigated whether such a relationship would also occur for other pro-inflammatory cytokines.

Despite the many studies conducted by researchers, for the time being, it is impossible to recognize cytokines as biomarkers for assessing the health of athletes, which would be useful for trainers, coaches, and personal trainers. This is due to the lack of reference values to which the results obtained could be related and considered elevated or reduced. At this point, the best solution would be to take resting measurements and repeat them immediately after training or after a several-week/month training program. This approach makes it possible to determine individual reference values [19]. This seems to be the most reasonable solution, as cytokine levels can depend on many factors, such as age, gender, fitness level, intensity and duration of exercise, and the presence of underlying medical conditions.

Several studies suggest that aging is associated with an increase in inflammatory cytokines. A comparison of cytokine levels in healthy elderly (65–80 years old) and young people showed a 50% greater TNFα response to Streptococcus epidermis in the elderly [120]. Similar results were obtained in mitogen-stimulated culture supernatants, where elderly subjects (>74 years old) had significantly elevated levels of IL-6, TNFα, and IL-1β compared to young subjects [121]. Sex differences in cytokine levels have also been reported, with women generally exhibiting higher levels of pro-inflammatory cytokines compared to men [122]. This may be due to the fact that estrogen, a hormone that is more abundant in women, has been shown to increase cytokine production [123]. Genetics can also play a role in cytokine levels, with certain gene variations associated with increased cytokine production or decreased cytokine regulation [124]. Lifestyle factors such as diet and exercise can also impact cytokine levels. A diet high in processed foods and saturated fats has been shown to increase levels of pro-inflammatory cytokines, while a diet rich in fruits, vegetables, and whole grains can help reduce inflammation [125]. Exercise can also impact cytokine levels, with acute exercise typically resulting in a transient increase in pro-inflammatory cytokines followed by a period of anti-inflammatory cytokine production. Fitness level is another important factor that influences cytokine response to exercise. Individuals who are more physically fit tend to have a more muted cytokine response to exercise compared to those who are less fit. This may be due to the fact that regular exercise results in a chronic low-grade inflammation that primes the immune system and reduces the magnitude of the cytokine response to subsequent exercise. The intensity and duration of exercise also play a role in cytokine response variability. High-intensity exercise has been shown to result in a more robust cytokine response compared to moderate-intensity exercise, while longer durations of exercise tend to result in a more sustained cytokine response [23]. Finally, the presence of underlying health conditions can also impact cytokine response to exercise. For example, individuals with chronic inflammatory conditions such as rheumatoid arthritis or inflammatory bowel disease may have a dysregulated cytokine response to exercise that differs from that of healthy individuals [126].

An attempt to identify unique cytokine profiles for different sports was made by Sohail et al. [127]. In their study, among the selected cytokines, they observed that certain sports were characterized by characteristic increases or decreases in the concentration of particular cytokines. In their analysis, they included disciplines, such as aquatics, athletics, cycling, football, and weightlifting. Among the cytokines studied, they observed that only MIP-1β was higher in weightlifters and football players compared to athletics and aquatics. In addition, they noted characteristically lower levels of HGF in aquatic sports compared to the other disciplines. IL-6 concentrations were significantly higher in soccer players than in aquatic sports, athletics, cycling, or weightlifting. IL-12 concentrations were significantly lower in aquatic sports than in the other disciplines.

Continued research into the use of cytokines as biomarkers in sports and exercise can bring many benefits, such as improved monitoring of training responses, early detection of overtraining and injury risk, personalized training programs, objective evaluation of training interventions, and a better understanding of physiological responses to exercise. However, in order to prepare a universal methodology for measuring cytokines to analyze the health status of athletes, it is necessary to standardize testing. As of today, several methods exist to measure cytokine levels in biological samples, including blood, serum, plasma, urine, and tissue (for example, muscle biopsies or tissue analysis in animal model experiments), which are currently used in scientific papers [128]. One of the most commonly used methods is the enzyme-linked immunosorbent assay (ELISA), which detects cytokines by using antibodies that are specific to each cytokine. Another method for measuring cytokine levels is multiplex cytokine assays, which allow for the simultaneous measurement of multiple cytokines in a single sample. Finally, molecular biology techniques, such as reverse transcription polymerase chain reaction (RT-PCR) and gene expression arrays, can also be used to measure cytokine levels. These methods measure the mRNA levels of cytokines, which provides information about the gene expression of the cytokine rather than the protein level [128,129]. Using so many techniques to measure the same parameters can result in differences in measurements. 

The second aspect worth noting when considering the use of cytokines as biomarkers in sports is the accounting for different sport disciplines. Each sport is characterized by different intensities, and as Sohail et al.’s research has shown, these can be characterized by different changes in the concentration values of individual cytokines. As we have already shown, the measurement of single cytokines does not provide conclusive information on the health status of an athlete; however, reducing the number of parameters tested will enable cost savings and may contribute to their easier application as routine tests for athletes.

We pointed to the lack of reference values to which test results could be compared as the main difficulty in using cytokines as biomarkers. Currently, resting values obtained from samples taken under training are used to compare results. This approach may have positive effects that allow each athlete to be treated on an individual basis. However, future research should focus on determining what increases and decreases in concentration of particular cytokines can be considered as significant and helpful in planning further training. 

While the potential for personalized training programs based on cytokine responses is an exciting area of research, there are still many unanswered questions and challenges that need to be addressed before this approach can be widely adopted. These include the development of standardized protocols for cytokine measurement, the identification of reliable cytokine biomarkers for specific training outcomes, and the integration of cytokine monitoring into practical training programs.

## 5. Conclusions

Recent research has suggested that the use of cytokine as biomarkers could enable the development of personalized training programs tailored to individual responses to exercise. The traditional approach to training has been to prescribe standardized exercise programs to all individuals based on their age, fitness level, and training goals. However, this approach does not account for the fact that individuals can respond differently to the same exercise stimulus, and that these responses can be influenced by factors, such as genetics, lifestyle, and previous training history. By monitoring cytokine responses to exercise, it may be possible to identify individual variations in the inflammatory and immune responses to exercise and to use this information to tailor exercise programs to maximize the benefits of training while minimizing the risk of injury and overtraining.

For example, individuals who demonstrate a higher level of cytokine response to exercise may require longer recovery times between sessions or lower overall training volumes, while those with a lower cytokine response may be able to tolerate more frequent or higher-intensity training. Furthermore, the use of cytokine biomarkers could enable the development of more targeted interventions to promote recovery and adaptation to training, such as the use of anti-inflammatory interventions for individuals with a high cytokine response or the use of immune-boosting interventions for those with a low cytokine response. However, due to the impossibility of defining reference values for individual cytokines that were universal for all, we conclude that the basis for using cytokines as biomarkers should be at least two measurements. The first should take place before the start of an exercise session to obtain a resting value, and the second after the exercise is completed. At this point in time, where a lot of new work is being produced that gives new insights into the contribution of cytokines to exercise, it is very difficult to propose one specific panel of cytokines that could be used today. Based on studies that compare trained and untrained individuals, we suggest that the use of cytokines, such as IL-1β, IL-6, TNF-α, IFN-γ, and IL-1ra, may be useful for assessing the body’s adaptation to training and the body’s increase in training. However, at this stage in the development of this field of science, we do not want to exclude the use of other cytokines, such as chemokines or growth factors, which also appear to be promising indicators, but need to be studied more extensively. At present, the most cost-effective method for performing the studies we suggest seems to be reverse transcription polymerase chain reaction and gene expression measurement using Syber Green.

In conclusion, the future use of cytokines as biomarkers to assess training effectiveness (especially in professional athletes) may offer several potential benefits. However, for this to happen, larger studies that include both athlete and non-training groups are needed. In addition, by continuing research in this direction, we have the opportunity to better understand the physiological responses to exercise and how they differ between individuals. This could lead to new insights into the effects of exercise on the body, new strategies for optimizing training programs and injury prevention, and the use of physical activity for disease prevention.

## Figures and Tables

**Figure 1 ijms-24-11156-f001:**
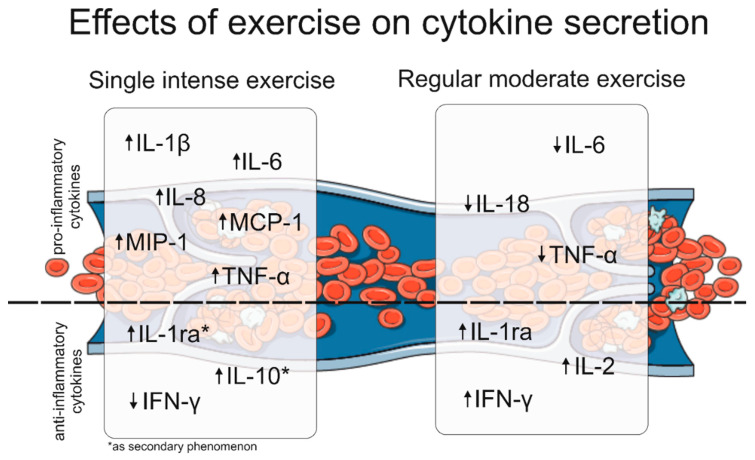
Graphic representation of the effect of single intense exercise and regular moderate exercise on plasma pro- and anti-inflammatory cytokine secretion levels (↑—increased secretion of a particular cytokine; ↓—decreased secretion of a particular cytokine). This figure was partly generated using Servier Medical Art, provided by Servier, licensed under a Creative Commons Attribution 3.0 unported license.

**Figure 2 ijms-24-11156-f002:**
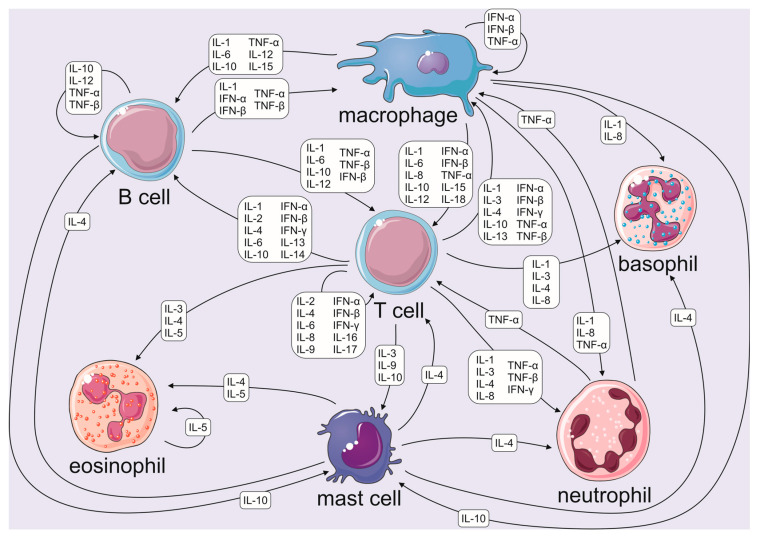
Diagram showing the different types of immune cells and how they interact with each other via cytokines. This figure was partly generated using Servier Medical Art, provided by Servier, licensed under a Creative Commons Attribution 3.0 unported license (based on [20]).

**Table 1 ijms-24-11156-t001:** Effects of exercise on interleukins secretion in humans.

Cytokine	Physical Activity	Training Level of the Subjects	Response	References
IL-1β	Running 20 km	Trained runners	Significantly increased levels of IL-1β are found 1 h after running and persist until the next day	[26]
IL-1ra	Treadmill test	Trained runners	Increased level of IL-1ra by 7.8–33% after test and 1 h after the test it dropped significantly	[27]
	Ultramarathon 160 km	Trained runners	Increased level of IL-1ra 6.1-fold after ultramarathon	[28]
	Cycling ergometer tests	Trained endurance athletes	Increased level of IL-1ra after 1 h of recovery, where it reaches a peak concentration	[29]
IL-18	Cycling 230 km	Trained cyclists	Significantly lower levels of IL-18 24 h after test	[30]
	Aerobic exercise training	Untrained men and women	Decreased level of IL-18 by 43% after training	[31]
	Exercise training	Obese individuals	Decreased IL-18 mRNA by 20% in adipose tissue after training	[32]
IL-2	Running 5 km	Trained runners	Significantly decreased IL-2 concentrations directly after race and significantly increased after 24 h	[34]
	Marathon 42 km	Trained runners	Decreased IL-2 concentrations by 32% or 55% directly after the race	[35,36]
	6-week program of aerobic dance exercise	Untrained women	Significantly increased level of IL-2 in after training program	[37]
	2 km rowing ergometer test	Trained rowers	Significant (about 4-fold) increased level of IL-2 after test and a return to normal after 24 h	[38]
IL-4	Running	Trained runners	No significant differences in IL-4 levels	[35,36,39,40,41]
	Rowing ergometer test	Trained rowers	No significant differences in IL-4 levels	[42]
	Running 40 km	Trained runners	No significant differences in IL-4 levels	[43]
	Running 171 km	Trained runners	Significantly increased level of IL-4 after running	[43]
IL-6	Marathon 42 km	Trained runners	Increased level of IL-6 45-fold after running and 1 h afterward	[49]
	Marathon 42 km	Trained runners	Increased level of IL-6 100-fold after running	[35]
	Ultramarathon 160 km	Trained runners	Increased level of IL-6 50.2-fold after running	[28]
	1 h rowing ergometer test	Trained rowers	Significantly increased mean level of IL-6 after test and after a 30 min rest	[42]
	90-min session on the ergometer	Trained rowers	Increased level of IL-6 7.5-fold 2 and 3 h after the test	[50]
	4 months of dance training	Untrained women	Decreased resting IL-6 concentrations by 60% after training program	[51]
	6-week program on a bicycle ergometer	Untrained men	No significant differences in IL-6 levels	[52]
IL-10	Ultramarathon 160 km	Trained runners	Increased level of IL-10 9.5-fold after running	[28]
	Marathon 42 km	Trained runners	Increased level of IL-10 3.5-fold after running	[35]
	1 h rowing ergometer test	Trained rowers	Non-significantly reduced level of IL-10 after test and after a 30 min rest	[42]
IL-12	Treadmill test	Trained runners	Significantly increased level of IL-12 p40 after a test	[66]

**Table 2 ijms-24-11156-t002:** Effects of exercise on chemokines secretion in humans.

Cytokine	Physical Activity	Training Level of the Subjects	Response	References
IL-8	Marathon 42 km	Trained runners	Increased level of IL-8 6.7-fold 30 min after running	[72]
	Ultramarathon 160 km	Trained runners	Increased level of IL-8 2.5-fold after running	[28]
	1 h cycling	Trained cyclists	Significantly increased level of IL-8 after cycling	[73]
	1 h cycling	Trained cyclists	30-fold increased IL-8 mRNA in skeletal muscles immediately after exercise	[73]
MCP-1	1 h rowing ergometer test	Trained rowers	Significantly increased level of MCP-1 after test	[42]
	Marathon 42 km	Trained runners	Significantly increased level of MCP-1 after running	[74]
	Wingate Anerobic Test	Trained soccer players	Minimal MCP-1 mRNA expression after the test and continuous increase over time with a maximum at the sixth hour after test	[75]
	Wingate Anerobic Test	Untrained individuals	Significantly increased MCP-1 mRNA expression immediately after the test and then decreased to a lower level	[75]
	Treadmill test	Trained runners	Significantly increased level of MCP-1 after running	[66]
MIP-1α	Marathon 42 km	Trained runners	Increased level of MIP-1α 3.5-fold 30 min after running	[72]
MIP-1β	Marathon 42 km	Trained runners	Increased level of MIP-1β 4.1-fold 30 min after running	[72]

## Data Availability

Data sharing is not applicable.

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
