# Peer review of "Cytokines as Biomarkers for Evaluating Physical Exercise in Trained and Non-Trained Individuals: A Narrative Review"

_ijms, 2023, doi:10.3390/ijms241311156_

Round 1

Reviewer 1 Report

Thank you for the invitation to review “Cytokines as biomarkers for evaluating physical exercise in trained and non-trained individuals – narrative review”. This review article by MaÅ‚kowska and Sawczuk aims to provide an overview of cytokines, their potential as biomarkers, methods for measuring cytokine levels, and factors that can affect cytokine levels. Nonetheless, the review is well structured and clearly written, but the authors have described a very broad topic of the scientific literature and in some respects the description is rather generic and not very thorough. As the effects of exercise are discussed, the authors should elaborate more on cytokines produced by skeletal muscle, such as IL-6.

Reviewer 2 Report

This review aims to give an overview on cytokines induced/related to physical activity and their potential to be used as biomarkers for evaluating physical exercise in trained and non-trained individuals with a focus is on pro versus anti-inflammatory cytokines. While overall this review is well written, it is rather descriptive and there are some points that, in my opinion, need to addressed in order to make it more comprehensive and useful in a practical way.

1. The selection of cytokines covered in this article is a bit surprising. There is a lengthy discussion on TNFalpha which is implicated in the obesity/diabetes related development of insulin resistance and certainly not a candidate for a biomarker associated with physical activity. On the other hand, growth factors in general are mentioned only shortly and there is no mention at all of FGF21 and GDF15 which are both upregulated by physical exercise and have been an intensive focus of research in the last few years; not only as metabolic factors but also as potential myokines and exercise  biomarkers. (For recent reviews see e.g. PMID: 37108444; 35250869; 34831213; 34554363; 33982072; 26898145, references to original articles can be found within these reviews). A timely and state of the art review must not omit these factors.

2. On page 13 methods for measurement of cytokines are shortly described. The descriptions are very superficial and, from a practical point of view not very useful because they present a mixture of methods for non-invasive, minimal invasive and very invasive approaches. Some, such as flow cytometry (does this refer to FACS sorting of immune cells?) and gene expression analysis in tissues (does this refer to muscle biopsies?) Are certainly not applicable as routine measurements outside of clinical studies. Also, most of these methods are very costly and can only be performed in specialized (research) laboratories.

3. In the conclusions the authors state: “By monitoring cytokine responses to exercise, it may be possible to identify individual variations in the inflammatory and immune responses to exercise, and to use this information to tailor exercise programs to maximize the benefits of training while minimizing the risk of injury and overtraining”. I find this statement also a bit unsatisfactory and lacking in more detailed suggestions how to implement this in praxis. Should this entail blood sampling at different timepoints before and after exercise? What time points? Which are the most promising cytokines to measure according to the current state of knowledge? Which methods would be most cost efficient to perform? Are these in any way realistic options for the normal population or rather something only available for professional athletes?

Round 2

Reviewer 1 Report

This second revision improved and completed the review. I accept this second version for publication in present form

Reviewer 2 Report

Thank you for the revision which, in my opinion, has greatly improved this review article.